# Spatial Patterns and Determinants of Bed and Breakfasts in the All-for-One Tourism Demonstration Area of China: A Perspective on Urban–Rural Differences

Ao Sun [1,2], Lin Chen [1,2,*], Kunimitsu Yoshida [3] and Meng Qu [4]

1 School of Geography, South China Normal University, Guangzhou 510631, China; ray_suen@m.scnu.edu.cn
2 The Center for Asian Geographical Studies, South China Normal University, Guangzhou 510631, China
3 Faculty of Geo-Environmental Science, Rissho University, Kumagaya 360-0194, Japan; ysh-9232@ris.ac.jp
4 Center for Advanced Tourism Studies, Hokkaido University, Sapporo 060-0817, Japan; meng@cats.hokudai.ac.jp
* Correspondence: chenlin.geo@m.scnu.edu.cn

**Abstract:** The spatial structure of Bed and Breakfast (B&B) development plays a crucial role in promoting integrated urban–rural development. However, existing B&B research has predominantly focused on single large cities, neglecting to explore the spatial patterns of B&B development and their influencing factors from the perspective of urban–rural differences. To address this gap, we conducted a comprehensive case study in an all-for-one tourism demonstration area in Hainan Province, China. We adopt geospatial analysis methods and ridge regression models to investigate the characteristics of urban–rural disparities in B&B distribution and to identify the primary factors influencing their spatial arrangement. The research findings reveal valuable insights: (1) B&B establishments in the tourism demonstration area exhibit clustering with notable variations in clustering intensity between urban and rural regions; (2) essential factors affecting the spatial distribution of B&Bs include transportation accessibility, reliance on tourism attractions, B&B development infrastructure, and the availability of living services; (3) tourism resource dependence emerges as the most significant driving force behind B&B agglomerations in the tourism demonstration area; and (4) road network density, hotel service availability, and neighborhood residential density are three additional critical factors affecting B&B distribution, with their influence varying between urban and rural B&Bs. Based on these key findings, we propose development strategies for optimizing B&Bs' spatial structure in the tourism demonstration area and outline a blueprint for fostering integrated urban–rural development.

**Keywords:** B&B; all-for-one tourism demonstration area; urban–rural difference; ridge regression analysis; Hainan province; China





## 1. Introduction

Since the rapid development of urbanization and industrialization in China under the reform and opening-up policy, the issues of unbalanced urban–rural growth and inadequate rural development have become increasingly prominent. Accelerating the pace of the integrated development of urban and rural areas is a meaningful approach to realizing rural revitalization. All-for-one tourism is a development model that emphasizes the creation of comprehensive tourism resources, encompassing both cities and villages, along with their supporting facilities, to promote the coordinated development of regional society and economy.

Accommodation, as a vital tourism component, provides insights into the level of regional tourism development. The burgeoning B&B, representing a novel avenue of tourism development, has garnered escalating scholarly attention. Originating in Europe, the B&B concept has reached a state of maturity in Japan. In Western contexts, B&Bs epitomize rural sojourns, wherein local inhabitants extend their residences to tourists,

affording them proximity to nature and immersive engagement in culturally infused rural activities. China, in contrast, conceptualizes B&Bs as modest-scale lodging establishments harnessing latent local resources, facilitated by proprietors who actively participate in delivering an indigenous experience encompassing regional ecology, cultural production, and lifestyle. Diverging from Western paradigms, Chinese B&Bs pivot towards offering not only accommodation but a distinct way of life, emphasizing environmental integration, leisurely ambiance, and a departure from conventional rural living. The ascendancy of B&Bs is intrinsically tied to the flourishing sharing economy and digital innovations, allowing for their proliferation across urban landscapes and expansive rural domains [1,2]. Therefore, B&Bs offer a more accurate reflection of the actual state of regional rural tourism than traditional accommodation options.

As B&Bs have witnessed a rapid increase in number and gained popularity among tourists, researchers have focused on studying their impact on local social development. On the one hand, B&Bs' development has positive effects on regional socioeconomic growth, such as increasing residents' income, reducing the cost burden for tourists, easing dependence on traditional accommodation facilities, and lowering transportation costs between tourist destinations [3–5]. They also expand the tourism market and provide new employment opportunities, thereby fostering local development [6], among other benefits. On the other hand, B&Bs' development can also lead to adverse effects on regional socioeconomic growth, including resident displacement and gentrification, competition with traditional accommodation sectors, rising housing costs, noise and waste management issues, and conflicts between residents and tourists [7–9].

Second, the B&Bs tend to focus on cities and areas with abundant tourism resources. Studies on B&B distribution patterns in different cities, such as New York, Los Angeles, and Chicago, have revealed a core–edge spatial distribution pattern of urban B&Bs [10]. Additionally, B&Bs cluster around places rich in tourism resources such as resorts [11–15]. Research on Bulgarian B&Bs also highlights their concentration around commercial, upscale, and officially certified attractions [16]. Studies in Chinese cities, such as Nanjing and Suzhou, suggest that B&Bs are clustered in urban centers and old towns with abundant historical and cultural resources [17,18]. The development of tourism B&Bs is influenced by various factors, including tourism resources and their supporting facilities as well as the primary conditions for B&B development. Tourism resource locations as tourist destinations are central to attracting tourists. Tourism resources and their supporting facilities influence the distribution of B&Bs such as art, human landscapes, and nightlife spots [14]. Moreover, factors such as the number of amenities, business apartments [18,19], and population size [20] also affect B&B distribution.

Existing B&B-related research has focused on various spatial scales, ranging from national [11] to single-city analyses [10,12,14,15,17,19,21,22]. However, it is essential to focus on regional B&B development beyond large cities. Given the urban–rural differences in B&Bs as a service industry for urban consumers, it is crucial to explore the development of regional B&Bs with urban–rural differentiation. The urban–rural dichotomy in China, influenced by the long-term dual-structure system, has led to disparities in public service facilities, employment opportunities, and industrial development [23–27], among others. This study aims to investigate the characteristics of urban–rural B&B development and its influencing factors within an all-for-one tourism demonstration area, answering three key research questions: the spatial clustering characteristics of B&Bs in urban and rural regions of the all-for-one tourism demonstration area, the main factors influencing B&B development in the region, and the differences between urban and rural areas regarding these influencing factors. The rest of this paper is organized as follows: Section 2 discusses the theoretical background, including clustering characteristics, the global perspective, and the urban–rural perspective. Section 3 provides an overview of the all-for-one tourism demonstration area, along with the methods for data collection and selection of the factors influencing B&B development. Section 4 identifies the characteristics of the urban and rural spatial distribution of B&Bs in the all-for-one tourism demonstration area and the

main influencing factors. Section 5 focuses on the differences in agglomeration patterns and influencing factors of urban and rural B&Bs. Finally, the paper summarizes the main conclusions and offers suggestions for the construction of an all-for-one tourism demonstration area.

## 2. Theoretical Background

As a significant component of tourism accommodations, B&Bs play a dual role in the tourism system by providing both tourism experiences and accommodation services, particularly in the context of sharing economy development. Spatial analyses have revealed a strong spatial relationship between B&Bs and other tourism products and supporting facilities, indicating spatial proximity and heterogeneity.

From a global perspective, B&B development is influenced by various factors, including transportation accessibility, the foundation of B&Bs, the availability of living service support, tourist attractions, and demographic conditions. However, the degree of influence of these factors varies widely and spans various aspects of tourism products and facilities. Better elucidating these relationships and capturing the spatial proximity of B&Bs requires identifying the differences in the impact of these factors from a global perspective. However, focusing solely on a global perspective may overlook spatial heterogeneity and fail to account for regional disparities.

To explore the relationship between B&Bs and their influencing factors more comprehensively, adopting an "urban–rural differences" perspective proves beneficial. This approach transcends the global perspective and acknowledges the discrepancies in the extent to which influencing factors affect accommodation distribution in urban and rural areas. The urban–rural perspective provides a more detailed spatial delineation, aligning with the actual conditions in China, and accommodating the marked urban–rural discrepancies in the B&B distribution. While traffic location, B&B development prerequisites, living service support, and tourist attractions remain influential factors from an urban–rural perspective, the mechanisms underlying their impact may differ. Consequently, the urban–rural perspective can address the gaps that emerge from a global perspective and rectify its limitations.

In summary, the relationship between the distribution of B&Bs and its influencing factors can be thoroughly explored from the theoretical perspectives outlined above. First, B&Bs' spatial distribution reflects the clustering characteristics of this new form of accommodation driven by the sharing economy and digital technology. Second, the global perspective highlights the significance of factors influencing B&B distribution, regardless of urban–rural considerations. Finally, the urban–rural perspective delves deeper into the influencing factors and their mechanisms in B&B distribution under urban–rural differences. This structured analytical framework serves as an essential reference for future research in the tourism industry and the study of urban–rural distinctions (Figure 1). In subsequent research, we can further deepen our understanding of B&B development and investigate the relationship between B&Bs and their influencing factors to promote sustainable tourism industry development and the coordinated growth of urban and rural areas.

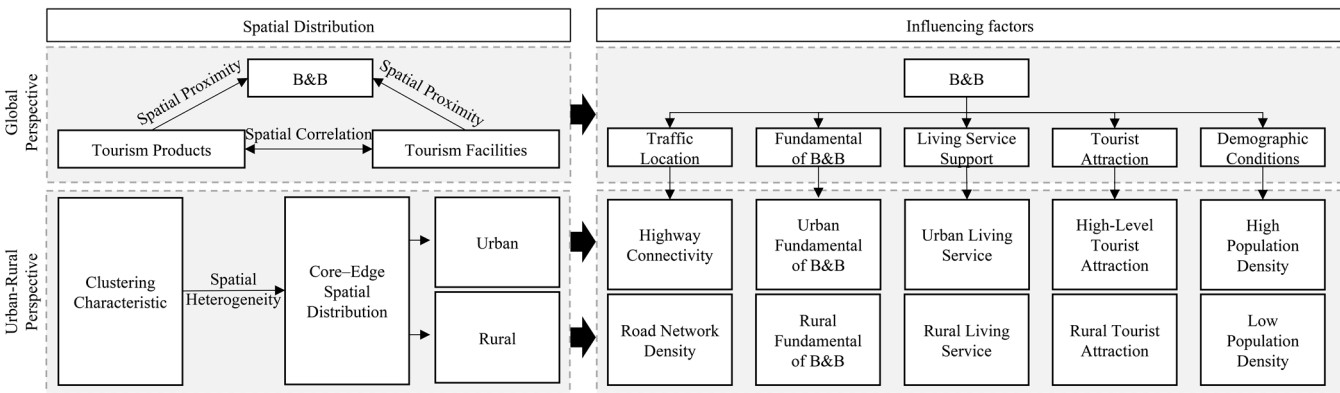

**Figure 1.** Analytical framework.

## 3. Material and Methods

### 3.1. Study Area

The establishment of all-for-one tourism demonstration areas in China aims to leverage tourism development to drive overall regional growth and bridge development disparities between regions. The concept of national all-for-one tourism demonstration areas was introduced in the Chinese government documents in 2015. However, in 2017, the China National Tourism Administration (CNTA) officially released guidelines for their establishment, marking the official initiation of these areas. Subsequently, in 2018, the General Office of the State Council released Guiding Opinions on Promoting the Development of Territorial Tourism, further supporting the creation of these demonstration areas. Following these developments, local governments in China have actively attempted to establish All-for-one Tourism Demonstration Areas.

One such example is the Hai-Cheng-Wen B&B service circle located in the northern part of Hainan Province, which encompasses Haikou City, Wenchang City, and Chengmai County (Figure 2). Within this arrangement, Haikou City functions as a prefecture-level jurisdiction consisting of four districts, while Wenchang City and Chengmai County are designated as county-level cities and counties, respectively. Precisely, Haikou City, included 21 streets and 22 towns within its administrative purview in 2022, Wenchang City encompassed 17 towns, and Chengmai County comprised 11 towns. These administrative divisions of streets, towns, and townships hold paramount significance in demarcating China's urban and rural territories. As established through prior methodological frameworks, streets (Jiedao) are classified as urban areas, while townships (Xiang) and towns (Zhen) are categorized as rural localities [28]. The Hai-Cheng-Wen B&B service circle thereby exemplifies a paradigmatic illustration of an integrated urban–rural developmental context, encapsulating both urban and rural components.

The development plan for rural B&Bs in Hainan (2018–2030) seizes the opportunity provided by the construction of the Hainan Free Trade Pilot Zone (port) and outlines the development of distinctive B&B projects. The plan emphasizes the exploration of local cultural resources, optimization of the spatial arrangement of B&Bs, and establishment of an area-wide linkage of rural B&B development patterns. The Hai-Cheng-Wen B&B service circle has been designated as a key area for building an international tourism island and free-trade port in Hainan Province. The area benefits from the prominent socio-economic status of Haikou, the provincial capital, which serves as a political, economic, and transportation center.

The 2021 Hainan Provincial Statistical Yearbook indicates that in 2020, the GNP of the Hai-Cheng-Wen B&B service circle accounted for 43.44% of Hainan's total value, and the number of overnight visitors received by tourist hotels accounted for 25.8% of the province's total number of visitors. However, there are substantial disparities in the B&B development between urban and rural areas within the demonstration zone because of uneven administrative, economic, and tourism resource allocation. As such,

this demonstration area was selected as the study site to investigate the main factors influencing the urban–rural differences in B&B development. By doing so, this research aims to provide scientific support for the integrated development of urban and rural areas and has significant research implications.

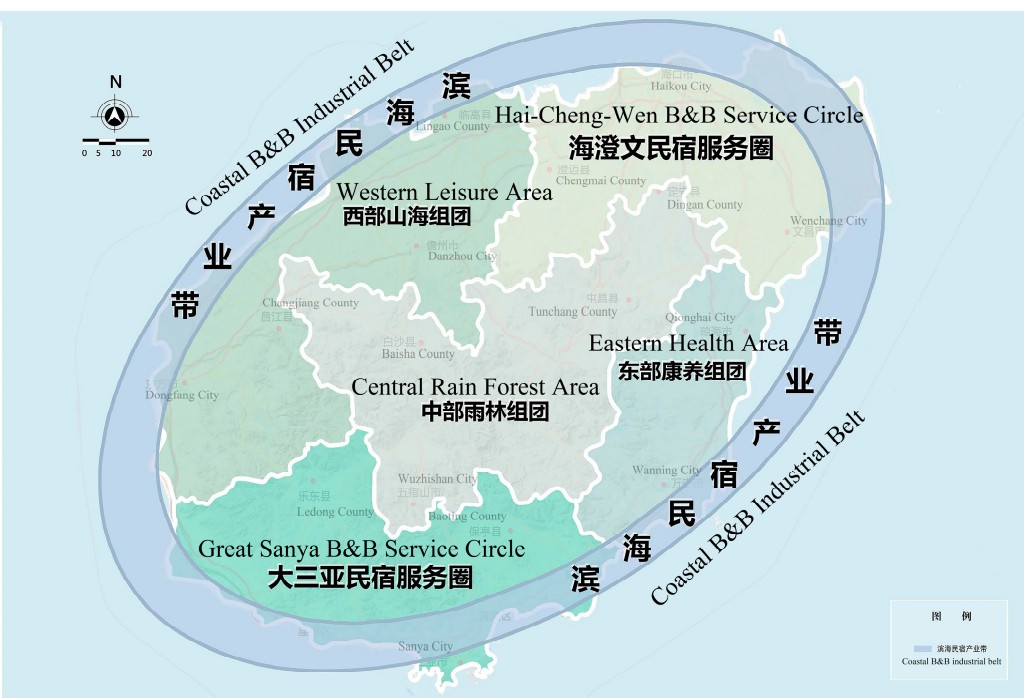

**Figure 2.** Spatial structure plan of rural B&B in Hainan Province.

*3.2. Data Collection*

B&Bs represent distinct forms of tourist hospitality facilities, distinguishing themselves from conventional hotels, often securing reservations through online platforms. Notably, Airbnb, a service-based website connecting travelers and hosts, was established in 2008 and entered the Chinese market in 2015, quickly becoming a prominent player in the B&B industry. The Internet B&B platform has been widely studied in the context of regional tourism development and its impact on the overall tourism experience in several research works [21,22,29–31].

In this study, web crawling techniques were employed to investigate the B&B landscape in the Hai-Cheng-Wen B&B service circle, which includes Haikou City, Wenchang City, and Chengmai County. The data obtained from the Airbnb website comprised essential information about the B&Bs, such as names, latitude and longitude coordinates, addresses, ratings, and registration times, which were then transformed into Points of Interest (POI) data points. To ensure data accuracy and validity, 1278 unique B&B listings were obtained by overlaying the geographic data of administrative divisions using ArcGIS software to remove data points with offset latitudes and longitudes and to eliminate duplicate entries.

To facilitate a comprehensive analysis of the differences in B&B development between urban and rural areas, this study utilized China's most fundamental administrative divisions, namely townships, towns, and streets, as the unit of analysis. Streets were classified as urban areas, whereas townships and towns were categorized as rural areas, based on a method previously employed in research [28]. According to this classification, within the demonstration area, 574 B&Bs were situated in urban areas and 704 were located in rural areas (Figure 3).

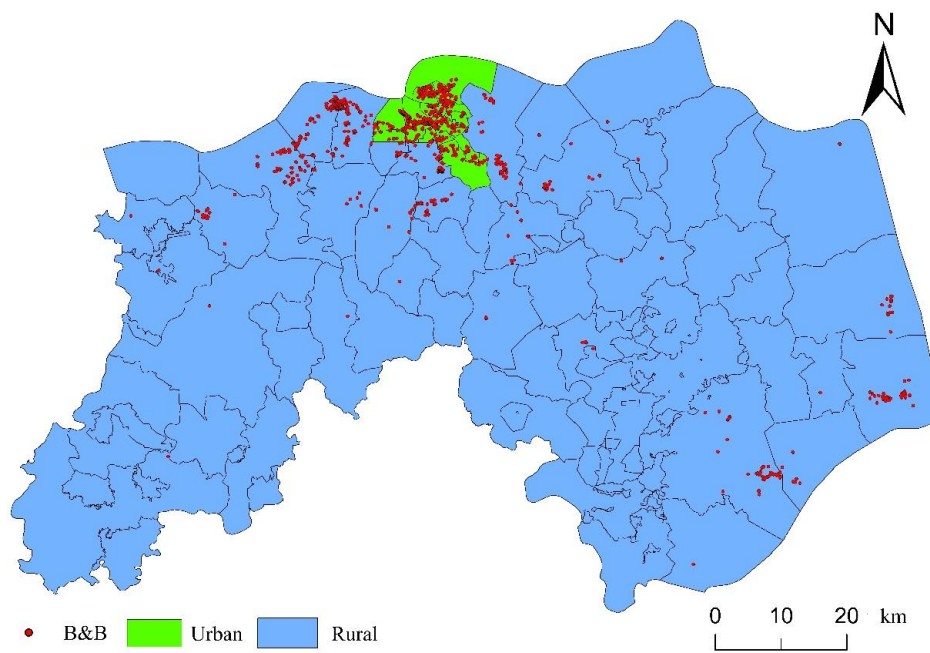

**Figure 3.** Spatial distribution of B&Bs in the demonstration area.

In addition to the B&B data, other data sources were used here (Table 1). Administrative divisions and road traffic network data were obtained from the China National Basic Geographic Information Center. Population data were sourced from the 7th National Population Census of China. The list of national A-class scenic spots and beautiful villages was obtained from the "List of national A-class scenic spots in the province" and the "Table of coconut-level rural tourism spots in Hainan Province" released by Hainan Province. The POI data for leisure and entertainment, hotel services, community residences, public facilities, and tourist attractions were sourced from the Baidu Map Search Service API. These data sources were thoroughly verified and provided reliable support for our research.

**Table 1.** Main sources of data acquisition.

| Data Name | Source | Acquisition Time |
|---|---|---|
| B&B information | Airbnb website: https://www.airbnb.cn/ | 20 January 2022 |
| Administrative divisions | China National Basic Geographic Information Center | 15 January 2022 |
| Traffic road network data | China National Basic Geographic Information Center | 15 January 2022 |
| National A-class scenic spots | Baidu Map Search Service API https://lbsyun.baidu.com/ | 2 December 2021 |
| Coconut-level rural tourism spots | Baidu Map Search Service API https://lbsyun.baidu.com/ | 2 December 2021 |
| Tourist attractions | Baidu Map Search Service API https://lbsyun.baidu.com/ | 2 December 2021 |
| Hotel services | Baidu Map Search Service API https://lbsyun.baidu.com/ | 2 December 2021 |
| Community residences | Baidu Map Search Service API https://lbsyun.baidu.com/ | 2 December 2021 |
| Leisure and entertainment | Baidu Map Search Service API https://lbsyun.baidu.com/ | 2 December 2021 |
| Public facilities | Baidu Map Search Service API https://lbsyun.baidu.com/ | 2 December 2021 |
| Population | The 7th National Population Census of China | 15 January 2022 |

### 3.3. Methods

3.3.1. Average Nearest Neighbor Index

The average nearest neighbor index (ANN) is widely used in geographic analysis to assess the spatial distribution patterns of point-like geographic phenomena. It is calculated by ranking the distances between all points and then determining the ratio between the average distance of each point to its nearest neighbor and the expected distance [32]. The index yields values between zero and infinity, where zero indicates a perfectly clustered spatial distribution of points, one indicates a random distribution, and values greater than one indicate a dispersed or uniform distribution. In the field of human geography, the mean nearest neighbor index is commonly employed to investigate various point-like spatial distribution types, including the arrangement of public service facilities [33], distribution of retail businesses [34], and the spatial layout of tourism products [35]. In this study, we used the average nearest neighbor index to analyze the spatial distribution patterns and characteristics of tourism B&Bs, which are a significant type of regional economic activity. This approach enables a rigorous examination of the spatial arrangement of the B&Bs and their implications for regional tourism development.

3.3.2. Kernel Density Estimation

Kernel density estimation is a widely employed spatial analysis technique that characterizes the intensity of geographic event distribution across a surface or mesh space [36]. This method involves interpolating point data onto a smooth density surface, thereby illustrating the magnitude of point density at various locations within the study area. By visually representing the spatial distribution of point data, kernel density estimation facilitates examining the degree of aggregation and regularity of the spatial distribution of human geographic phenomena in the area under investigation. Furthermore, this method can be used to compare point data distributions across different periods or spaces, enabling the identification of the evolutionary process of these phenomena. Consequently, kernel density estimation has extensive applications in diverse research fields, including urban planning, environmental management, and traffic analysis. In our analysis of tourism lodging and the POI data of various natures that impact its development, the kernel density estimation serves as a valuable tool for investigating geospatial distribution patterns. This technique enabled a thorough examination of spatial trends and correlations between tourism lodging and relevant POI data, shedding light on the spatial dynamics and potential influencing factors in the study area.

3.3.3. Spatial Autocorrelation

Spatial autocorrelation is a statistical test that examines whether the attribute values of geographic elements exhibit spatial correlation, indicating a tendency to cluster or disperse in geographic space [37]. There are two main categories of spatial autocorrelation: global and local. The widely used global spatial autocorrelation index is Moran's I, which quantifies the spatial correlation between each element and its neighboring elements within a region. The value of Moran's I ranges from $-1$ to 1: a positive value indicates spatial aggregation, a negative value signifies spatial dispersion, and a value close to 0 suggests a random spatial distribution.

Local spatial autocorrelation complements global spatial autocorrelation, and describes the spatial heterogeneity of different locations within a study area. It employs visualization tools such as Moran scatter plots and Local Indicators of Spatial Association (LISA) diagrams. The Moran scatter plot helps to detect spatial correlations among individual spatial units, revealing global spatial autocorrelation patterns. In contrast, the LISA plot illustrates local correlations among spatial units, highlighting local spatial autocorrelation patterns.

In this study, univariate and bivariate spatial autocorrelations were used to assess the degree of spatial agglomeration of B&Bs and their spatial relationships with other geographic elements. These analyses provide valuable insights into the spatial distribution patterns of B&Bs and their interactions with surrounding features, contributing to a com-

prehensive understanding of the spatial dynamics and potential influencing factors in the study area.

### 3.3.4. Ridge Regression Analysis

Ridge regression analysis is a valuable approach employed in situations in which covariant data exist. Covariance refers to strong correlations among independent variables that lead to variance inflation and an inaccurate estimation of regression coefficients when using the traditional least-squares method. In such cases, a ridge regression analysis provides a more realistic alternative, albeit at the expense of practicality and the unbiased nature of least squares [38]. The primary advantage of ridge regression is its ability to retain a significant portion of information and independence among independent variables, thus preserving the model's original explanatory power. By introducing a normal number matrix $kI$ to the singular matrix $X'X$, ridge regression analysis effectively mitigates matrix singularity issues, yielding more stable and accurate regression coefficients. The ridge-estimated coefficients are expressed as

$$\hat{\beta}(k) = \left(X'X + kI\right)^{-1}X'Y$$

where $\hat{\beta}(k)$ is the ridge estimator; when $k = 0$, it is the ordinary least squares estimate of OLS; and when $k \to \infty$, the ridge estimator tends to 0. A ridge plot often determines the value of $k$, and the principle of selection is the minimum value of $k$ when the standardized regression coefficient of each independent variable tends to be stable. In general, the smaller the value of $k$, the smaller the bias.

### 3.4. Selection of Variables Affecting B&B Development

Based on the prevailing trends in regional tourism development in Hainan Province and the specific characteristics of the current B&B industry, this study adopts the "B&B density" index as a metric to gauge the level of B&B development. The research included ten independent variables categorized into five groups (Table 2). In the process of discerning salient determinants shaping B&B development, a judicious selection framework has been applied. This has led to the inclusion of variables encompassing transportation accessibility, the degree of reliance on tourism attractions, the foundational bedrock for B&B proliferation, the provisioning of supportive residential amenities, and the demographic fabric. The selection criteria for these determinants are underpinned by an extensive engagement with antecedent research findings, thereby substantiating their significance. Concurrently, cognizant of the inherent variations between urban and rural areas within the spatial distribution of B&Bs and the underlying factors influencing this distribution, supplementary independent variables have been incorporated. These supplemental variables have been astutely chosen to elucidate and account for the nuanced urban–rural differentials within the determinants impacting B&B development dynamics.

First, with respect to transportation, two indicators, namely "road network density" and "distance from a highway" are selected. These indicators reflect the B&B industry's reliance on both intra- and inter-provincial transportation. Given the significance of transportation in facilitating tourism and B&B operations, areas with higher accessibility within cities are likely to be sought, whereas proximity to highways tends to attract out-of-town tourists.

Second, concerning tourist attraction, three indicators are chosen: "distance to national A-class scenic spots", "distance to a beautiful countryside", and "density of tourist attractions". These measures reflect the B&B industry's dependence on prominent tourist sites, high-quality rural attractions, and an overall abundance of surrounding tourism resources. In the tourism sector, attractions play a pivotal role in attracting tourists, and the location selection of B&Bs is often influenced by the distribution of these tourism resources.

**Table 2.** Variable selection and their meanings.

| Type | Name | Variables | Meanings | Calculation Method |
|---|---|---|---|---|
| / | B&B density | Y | The development level of B&B | Street, township B&B number/area |
| Traffic location | Road network density | X1 | City traffic accessibility | Grade road miles/area |
| | Distance from a highway | X2 | Inter-provincial traffic accessibility | Average closest distance of B&B from highway |
| Attraction dependence | Distance to national A-class scenic spots | X3 | The dependence of B&Bs on prominent tourist attractions | Average closest distance of B&B from national A-class scenic spots |
| | Distance to the beautiful countryside | X4 | The dependence of B&Bs on high-quality rural tourist attractions | Average closest distance of B&B from beautiful countryside |
| | Density of tourist attractions | X5 | The dependence of B&Bs on surrounding tourism resources | Number of tourist attractions/area |
| Foundation of B&B development | Hotel service density | X6 | The region's accommodation development level | Number of hotel services/area |
| | Neighborhood residential density | X7 | The region's housing support development level | Number of dwellings/area |
| Support of living services | Recreation density | X8 | The availability of recreational facilities | Number of recreational facilities/area |
| | Public service density | X9 | The availability of public services | Number of public services/area |
| Population conditions | Population density | X10 | The market base of the guest source in the region | Number of resident population/area |

Third, regarding the foundation of B&B development, two indicators are included: "hotel service density" and "neighborhood residential density". These indicators offer insight into the level of accommodation and housing support development in the region. A higher density of hotel services typically indicates a more mature and well-established lodging industry, whereas a greater concentration of neighborhood residences implies a more substantial housing supply. These factors significantly influenced the B&B industry's development.

Fourth, concerning the support of living services, the study incorporates the indicators of "recreation density" and "public service density". These measures reflect the availability of recreational facilities and public services in the area, which are essential considerations for the B&Bs.

Last, in terms of demographic conditions, the study includes the "population density" indicator. This measure provides an understanding of the market base for potential B&B guests in a region. Local residents often form a consistent consumer base in the context of the B&B development.

## 4. Results

### 4.1. Spatial Distribution Characteristics of B&Bs in the Demonstration Area

To explore the spatial distribution characteristics of B&Bs in the demonstration area, we conducted an analysis based on two types of data: B&B POI points and the number of B&Bs in each administrative division. Initially, we calculated the nearest neighbor index for the B&B POI data, resulting in a value of 0.2297, with Z = −52.67 and $p < 0.01$, indicating a clear clustering distribution pattern of B&B development in the demonstration area. Additionally, using kernel density analysis, we classified the B&B distribution into five density classes to identify hotspot areas with dense B&B concentrations. The findings revealed a "multiple core clusters and a coastal distribution" pattern with three primary B&B clusters: the Haikou urban B&B cluster, the Haikou-Chengmai B&B cluster, and the Wenchang B&B cluster (Figure 4). Notably, the Haikou urban cluster is part of a

metropolitan area, whereas the Haikou-Chengmai and Wenchang clusters are located in rural areas.

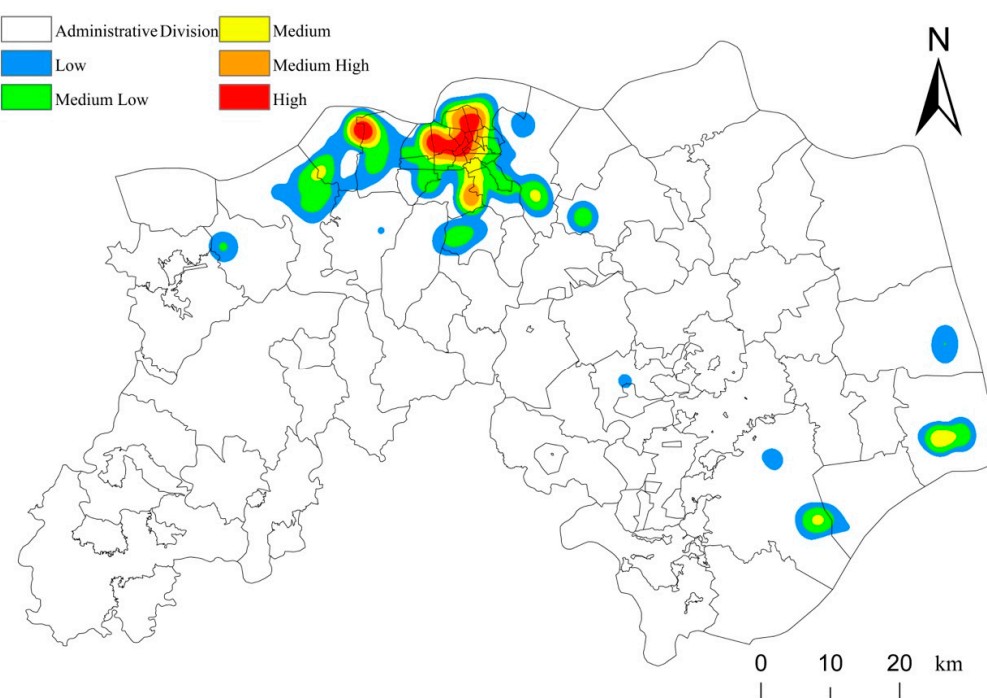

**Figure 4.** Kernel density distribution of B&Bs in the demonstration area.

Subsequently, we explored the spatial distribution characteristics by calculating the global and local spatial autocorrelation indices of B&B density using GeoDa software. The global autocorrelation index, represented by Moran's I value, measures spatial autocorrelation within the entire area. In our study, Moran's I value for B&B density was 0.736, which was significant, indicating a clear spatial clustering trend among B&Bs in the demonstration area.

However, the global autocorrelation index does not account for the spatial heterogeneity within the region. To address this, we employed a local spatial autocorrelation index (LISA) to investigate the spatial distribution differences within the demonstration area. The LISA classifies B&Bs into four types of spatial autocorrelations: high–high, high–low, low–low, and low–high. The results showed multiple highly clustered areas, with the high–high and high–low clusters primarily concentrated in the urban regions of Haikou City, whereas the low–low clusters were more prevalent in the non-coastal rural regions within Chengmai County and Wenchang City (Figure 5).

In summary, our study highlighted that the distribution of B&Bs in the demonstration area exhibited significant clustering and distinct spatial patterns. B&Bs tend to be concentrated in urban areas, with Haikou being a prominent hub and a region featuring prominent tourist attractions in Chengmai County and Wenchang City. In contrast, the B&Bs in most rural areas tend to be more dispersed, with relatively lower degrees of clustering. These findings shed light on the spatial dynamics of the B&B industry in the demonstration area and provide valuable insights for regional tourism planning and development in the Hainan Province.

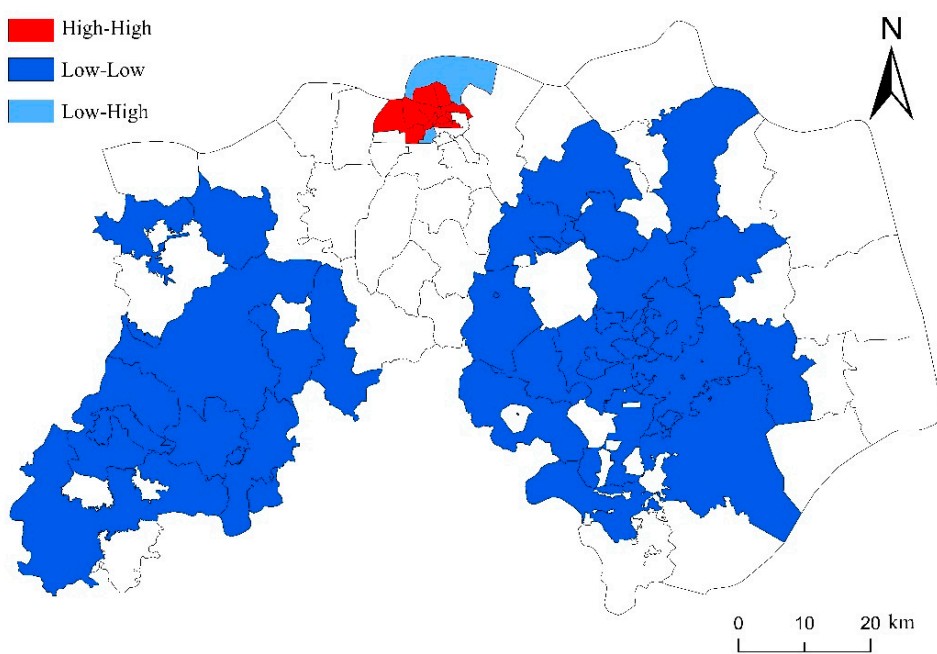

**Figure 5.** LISA cluster diagram of B&Bs in the demonstration area.

*4.2. Clustering Characteristics between B&Bs and the Factors Affecting Their Development*

The bivariate global Moran index presented in Table 3 provides significant insights into the spatial autocorrelation between B&Bs and various geographical factors, including transportation facilities, tourist attractions, hotel services, neighborhood housing, and public services. The results clearly indicate a strong association between the development of the B&B industry and the distribution of these factors.

**Table 3.** Bivariate global Moran's I statistics.

|  | B&Bs—Transportation Facilities | B&Bs—Tourist Attractions | B&Bs—Hospitality Services | B&Bs—Neighborhood Housing | B&Bs—Public Services |
| --- | --- | --- | --- | --- | --- |
| Moran's I Index | 0.716 | 0.483 | 0.645 | 0.705 | 0.650 |
| Z-Value | 12.092 *** | 9.399 *** | 11.380 *** | 12.019 *** | 11.417 *** |

Note: *** means 1% significance.

Among the geographical factors examined, the density of transportation facilities exhibited the strongest spatial correlation with the density of B&Bs, as indicated by a bivariate Moran's I index of 0.716 and a z-value of 12.092, both of which passed the 1% significance test. This suggests that areas with dense transportation facilities tend to have higher B&B concentrations. The distribution of transportation facilities played a pivotal role in fostering the development of the B&B industry.

Subsequently, the density of residential area subdivisions showed the next most substantial spatial correlation with B&B density, with a bivariate Moran's I index of 0.705 and a z-value of 12.019, passing the 1% significance test. This implies that areas with a high density of residential neighborhoods also tend to have a greater presence of B&Bs. The availability of housing options in the vicinity is likely to influence the B&B industry's growth.

Furthermore, the spatial correlations among public service density, hotel service density, tourist attraction density, and B&B density remain significant. The bivariate Moran's I indices for these factors were 0.650, 0.645, and 0.483 with corresponding z-values of 11.417, 11.380, and 9.399, respectively, passing the 1% significance test. These results underscore the relevance of public services, hotel services, and tourist attractions in shaping the B&B distribution.

*4.3. Factors Influencing the Spatial Distribution of B&Bs in the Demonstration Area*

4.3.1. Ridge Regression Model Selection and Interpretation

Collinearity refers to a high correlation between independent variables, which can lead to inaccurate and unstable results in the regression analysis. As shown in Table 4, we observed high collinearity in the models for all B&Bs, urban B&Bs, and rural B&Bs in the demonstration area, indicating strong correlations between the independent variables. For instance, the Variance Inflation Factor (VIF) values of road network density, tourist attraction density, hotel service density, neighborhood residential density, and public facility density in all three models exceeded 10, indicating a substantial degree of correlation. Such collinearity can pose challenges to the accuracy and stability of the regression analysis results.

**Table 4.** Colinear diagnosis results.

| Name of Variable | All B&Bs VIF | Urban B&Bs VIF | Rural B&Bs VIF |
|---|---|---|---|
| Road network density | 10.88 | 7.99 | 9.58 |
| Distance from a highway | 2.08 | 1.31 | 2.42 |
| Distance to national A-class scenic spots | 2.08 | 3.26 | 2.83 |
| Distance to the beautiful countryside | 1.87 | 3.31 | 2.64 |
| Density of tourist attractions | 8.85 | 8.73 | 10.37 |
| Hotel service density | 17.51 | 13.32 | 14.80 |
| Neighborhood residential density | 23.46 | 21.54 | 42.69 |
| Recreation density | 17.56 | 10.05 | 48.93 |
| Public service density | 22.33 | 17.30 | 10.52 |
| Population density | 13.11 | 6.84 | 24.11 |

To address the issue of independent variable collinearity, we opted for a ridge regression model to analyze the factors influencing B&B distribution. Table 5 presents the K-values of the ridge regression models for all B&Bs, urban B&Bs, and rural B&Bs in the demonstration area, as well as their corresponding $R^2$ and F-values. The $R^2$ values, which represent the goodness of fit of the models, are high for all three models, namely 0.798, 0.876, and 0.928, indicating that the models can effectively explain the observed data. Moreover, the F-values of all three models passed the 1% significance test, signifying the models' significance in explaining the dependent variable. These results demonstrate that the ridge regression model successfully analyzes the influencing factors affecting the spatial distribution of B&Bs in the demonstration area.

**Table 5.** Ridge regression model construction.

| | K | $R^2$ | F |
|---|---|---|---|
| all B&Bs | 0.184 | 0.798 | 15.419 *** |
| urban B&Bs | 0.174 | 0.876 | 7.059 *** |
| rural B&Bs | 0.168 | 0.928 | 23.328 *** |

Note: *** means 1% significance.

Using the ridge regression approach, we mitigated the impact of collinearity among the independent variables, thereby improving the accuracy and reliability of the regression analysis. The high $R^2$ values and significant F-values underscore the robustness of the ridge regression models in capturing the relationships between the dependent and independent variables, providing valuable insights into the factors that influence the distribution of B&Bs in the demonstration area.

4.3.2. Identifying the Factors Influencing the Distribution of B&Bs in the Demonstration Area

Based on the findings presented in Table 6, several factors were found to exert a significant influence on the distribution of all B&Bs in the demonstration area. The ridge

regression model analysis identified six variables that had a significant impact on the distribution of B&Bs: road network density, distance to national A-class scenic spots, hotel service density, neighborhood residential density, recreation density, and population density.

**Table 6.** All B&Bs ridge regression analysis results.

| Name of Variable | B | Std. | β | t | F |
|---|---|---|---|---|---|
| Constant | −0.06 | 0.526 | - | −0.114 | 0.910 |
| Road network density | 0.183 | 0.043 | 0.31 | 4.289 | 0.000 *** |
| Distance from a highway | 0.008 | 0.027 | 0.02 | 0.29 | 0.773 |
| Distance to national A-class scenic spots | −0.056 | 0.032 | −0.125 | −1.771 | 0.084 * |
| Distance to the beautiful countryside | 0.044 | 0.034 | 0.086 | 1.291 | 0.204 |
| Density of tourist attractions | 0.028 | 0.054 | 0.031 | 0.511 | 0.612 |
| Hotel service density | 0.079 | 0.031 | 0.165 | 2.571 | 0.014 ** |
| Neighborhood residential density | 0.045 | 0.013 | 0.197 | 3.412 | 0.002 *** |
| Recreation density | 0.068 | 0.017 | 0.264 | 4.093 | 0.000 *** |
| Public service density | 0.027 | 0.05 | 0.031 | 0.529 | 0.600 |
| Population density | 0 | 0 | −0.192 | −2.753 | 0.009 *** |

Note: ***, **, and * represent 1%, 5%, and 10% significance levels, respectively.

By assessing the significance and comparing the coefficients, we determined that road network density was one of the most critical factors influencing the distribution of B&Bs, as indicated by its significant coefficient. This suggests that B&Bs tend to concentrate in areas characterized by dense road networks, which can facilitate easy access for tourists and enhance the attractiveness of such locations for B&B development. Hotel service density emerged as another influential factor with a substantial coefficient, signifying that B&Bs in the demonstration area tended to cluster in regions with a higher density of existing hotels. This may be attributed to the B&B operators' preference to establish businesses in areas with a well-developed hospitality industry that could offer complementary services to guests and attract more tourists to the vicinity.

Moreover, the recreational density was found to exert a notable influence on the distribution of B&Bs, as evidenced by its relatively significant coefficient. This indicates that areas with a higher concentration of recreational facilities are more likely to attract B&Bs because these amenities can enhance the overall tourist experience and contribute to the appeal of locations for B&B guests.

### 4.3.3. Identification of Factors Influencing Urban B&B Distribution

Based on the predicted results of the ridge regression analysis presented in Table 7, several factors significantly influenced the distribution of urban B&Bs in the study area. Among these factors, distance from national A-class scenic spots emerged as one of the most crucial variables affecting the distribution of urban B&Bs. The regression results revealed that this variable exhibited the most significant influence coefficient and was the only one with a negative correlation. This indicates that the urban B&Bs in the demonstration area are generally situated at a considerable distance from the national A-class scenic spots.

The distance from a highway was also identified as a significant factor influencing the distribution of urban B&Bs, with its influence coefficient ranking second. Although the coefficient of the distance from a highway was relatively small, it remained statistically significant. This may be attributed to the Hainan Province's policy of free highway access, which has encouraged more urban B&B users to travel by car, leading to a greater demand for B&Bs in areas closer to highways.

Furthermore, hotel service density emerged as another influential factor in Urban B&B distribution, indicating that urban B&Bs in the demonstration areas tend to be concentrated in locations with a higher hotel density. This finding suggests that B&B operators strategically position their businesses in areas where hotels are already well-established, possibly leveraging the existing tourist infrastructure and attracting a larger pool of potential guests.

**Table 7.** Urban B&Bs ridge regression analysis results.

| Name of Variable | B | Std. | β | t | F |
|---|---|---|---|---|---|
| Constant | 2.225 | 2.884 | - | 0.772 | 0.458 |
| Road network density | 0.131 | 0.093 | 0.154 | 1.405 | 0.190 |
| Distance from a highway | 0.279 | 0.148 | 0.188 | 1.891 | 0.088 * |
| Distance to national A-lass scenic spots | −2.216 | 0.606 | −0.398 | −3.654 | 0.004 *** |
| Distance to the beautiful countryside | 0.163 | 0.152 | 0.105 | 1.067 | 0.311 |
| Density of tourist attractions | −0.045 | 0.064 | −0.067 | −0.702 | 0.499 |
| Hotel service density | 0.105 | 0.046 | 0.243 | 2.287 | 0.045 ** |
| Neighborhood residential density | 0.045 | 0.023 | 0.181 | 1.955 | 0.079 * |
| Recreation density | 0.07 | 0.026 | 0.292 | 2.677 | 0.023 ** |
| Public service density | 0.018 | 0.074 | 0.024 | 0.25 | 0.808 |
| Population density | 0 | 0 | −0.391 | −3.278 | 0.008 *** |

Note: ***, **, and * represent 1%, 5%, and 10% significance levels, respectively.

The presence of recreational facilities, measured by recreational density, also influenced the distribution of urban B&Bs, although the impact coefficients were relatively small. Nevertheless, this variable remained statistically significant, indicating that urban B&Bs tend to cluster in areas with a higher concentration of recreational amenities. This phenomenon can be attributed to the fact that the availability of recreational facilities can enhance a location's overall appeal for B&B guests.

Finally, neighborhood residential density was identified as another factor influencing the distribution of urban B&Bs, with a relatively small but significant influence coefficient. This suggests that urban B&Bs are more likely to be found in areas characterized by a higher density of residential neighborhoods.

4.3.4. Identification of Factors Influencing Rural B&B Distribution

Based on the ridge regression results presented in Table 8, the distribution of rural B&Bs in the demonstration area was influenced by various factors, among which five indicators passed the significance test: road network density, tourist attraction density, hotel service density, neighborhood residential density, and recreation density. These factors play crucial roles in shaping the spatial distribution of rural B&Bs and their implications are given below.

**Table 8.** Rural B&Bs ridge regression analysis results.

| Name of Variable | B | Std. | β | t | F |
|---|---|---|---|---|---|
| Constant | −0.101 | 0.085 | - | −1.188 | 0.250 |
| Road network density | 0.049 | 0.027 | 0.122 | 1.808 | 0.0878 * |
| Distance from a highway | 0.005 | 0.003 | 0.088 | 1.404 | 0.177 |
| Distance to national A -lass scenic spots | 0.002 | 0.004 | −0.103 | −1.677 | 0.111 |
| Distance to the beautiful countryside | −0.007 | 0.004 | 0.03 | 0.491 | 0.629 |
| Density of tourist attractions | 1.703 | 0.243 | 0.452 | 7.003 | 0.000 *** |
| Hotel service density | 0.258 | 0.188 | 0.14 | 2.193 | 0.042 ** |
| Neighborhood residential density | 0.155 | 0.041 | 0.203 | 3.724 | 0.002 *** |
| Recreation density | 0.068 | 0.024 | 0.105 | 2.805 | 0.012 ** |
| Public service density | −0.094 | 0.094 | −0.059 | −1.001 | 0.330 |
| Population density | 0 | 0 | −0.024 | −0.553 | 0.587 |

Note: ***, **, and * represent 1%, 5%, and 10% significance levels, respectively.

First, the density of tourist attractions emerged as one of the most critical factors affecting rural B&B distribution. A higher density of tourist attractions in a particular area can attract more tourists, thereby increasing the visibility and reputation of the region and promoting rural B&B development. Because tourists seek proximity to popular attractions, the B&Bs strategically position themselves in areas with higher concentrations of tourist hotspots.

Second, hotel service density is identified as a significant factor influencing the distribution of rural B&Bs. In some cases, tourists may prefer traditional hotel accommodations to rural B&Bs. Therefore, a higher hotel service density in a region may exert competitive pressure on the development of rural B&Bs. Rural B&Bs operators must consider this aspect when positioning and marketing their establishments.

Third, neighborhood residential density has emerged as another essential factor that affects the distribution of rural B&Bs. A higher density of residential neighborhoods suggests a relatively higher level of urbanization in the area. This urbanization process often leads to the conversion of residential properties into B&Bs. Consequently, a higher neighborhood residential density positively impacts the development and availability of rural B&Bs in the region.

Moreover, the density of recreational facilities plays a significant role in influencing rural B&B distribution. Tourists in rural areas often seek opportunities to immerse themselves in peaceful countryside settings while engaging in various recreational activities. Therefore, a higher density of leisure and recreational facilities enhances the overall appeal of the area to tourists, thereby promoting the development of rural B&Bs.

Finally, road network density emerged as one of the most crucial factors affecting the distribution of rural B&Bs. In rural areas, the availability and quality of transportation infrastructure significantly influence the convenience of tourists traveling to rural B&Bs. Hence, a well-developed and comprehensive road network can attract tourists to rural areas, thereby promoting the development and accessibility of rural B&Bs.

## 5. Discussion

### 5.1. Spatial Clustering Characteristics of B&Bs

From a spatial distribution perspective, there was a clear agglomeration of B&Bs in the demonstration area. Using kernel density analysis and statistics on urban and rural B&B numbers, we observe that although the total number of urban B&Bs is not as large as that of rural B&Bs, they exhibit a high distribution density, forming significant urban B&B agglomeration. By contrast, the distribution of rural B&B resources is relatively uneven, with most rural areas developing into large-scale B&B agglomerations.

Second, the local spatial autocorrelation analysis highlighted a more pronounced B&B agglomeration in the demonstration area's urban areas. A higher level of economic development and urbanization in cities contributes to an increased likelihood of B&B agglomeration. Additionally, there is a notable gap in B&B agglomeration levels between urban and rural areas. This phenomenon may be attributed to the fact that the tourism industry, as a service-oriented tertiary sector, primarily caters to high-income groups in urban settings, aligning with the spatial distribution trends of B&Bs observed in other countries [39].

Third, the clustering of B&Bs demonstrates a close interdependence with the distribution of other establishments. Bivariate local spatial autocorrelation analysis revealed that urban centers significantly influenced the spatial distribution of B&Bs. This is because of the concentration of transport facilities, residential areas, public services, and hotel services in urban regions, which consequently impacts B&B distribution. Although tourist attractions are also relevant to the level of the B&B agglomeration, their influence is relatively low. Hence, the disparity between urban and rural areas emerges as a crucial factor influencing the distribution of B&Bs. Additionally, the development of B&Bs may display the Matthew effect [40], wherein successful entities experience increased success, while weaker entities face greater challenges, even within the context of all-for-one tourism demonstration areas, which advocate integrated urban and rural development.

### 5.2. Global Perspective: Factors Influencing the Spatial Patterns of B&B

The spatial distribution of B&Bs holds significant importance within the tourism industry, as it influences tourism market development and enhances the overall tourism experience. In this study, we analyzed the factors influencing the spatial distribution of

B&Bs in the demonstration areas of Hainan Province, China, and identified five main influencing factors.

First, traffic location is a crucial factor that shapes the distribution of B&Bs. The accessibility of B&Bs is closely tied to traffic connectivity, and the well-developed road network in urban areas, driven by initiatives such as the Hainan International Tourism Island and the Hainan Free Trade Zone, has facilitated B&B establishment and accessibility. Consequently, the disparity in B&B accessibility among different regions within the demonstration area is relatively minimal.

Second, the fundamental factors underpinning B&B development, such as hotel service density and community residence density, also significantly impact B&B distribution. B&Bs tend to be located in neighborhoods that offer well-developed infrastructural support and provide an authentic experience to guests. This observation aligns with a study by Paulauskaite Dominyka, which highlighted that Airbnb users prefer exposure to local culture for authentic experiences [41].

Third, living service support factors, represented by recreational density, exert a notable influence on B&B distributions. In addition to tourism services, the availability of daily life products and support services, such as food and beverages, culture, entertainment, recreation, and retail offerings, play a pivotal role in shaping the vibrancy of neighborhoods [42,43] and consequently affect B&B distribution.

Moreover, tourist attractions also contribute to the B&Bs' distribution, albeit with a relatively smaller impact. This is attributed to two reasons. First, the broader spatial scale and larger area of this study compared to other research on urban accommodation and influencing factors led to a less pronounced impact from a global perspective. Second, proximity to scenic spots is no longer the sole determining factor because of the influence of transportation and other elements. Tourists may choose to stay in B&Bs far from national A-class scenic spots if better accommodations are available. This aligns with our calculation results, demonstrating tourist preferences for enhanced lodging experiences.

Finally, demographic conditions may have indirect effects on the B&B distribution. Areas with a higher population density indicate a more active consumer market, potentially attracting more B&B owners to open establishments. Moreover, local residents influence the operation and management of B&Bs, as they may have higher demands and expectations regarding the quality and services provided. Consequently, demographic conditions, while not directly impacting B&B distribution, warrant attention and exploration because of their potential impacts on B&B operations.

*5.3. Urban–Rural Perspective: Factors Influencing B&B Spatial Patterns*

The influence of transportation accessibility on B&B distribution in the demonstration area was noteworthy. With the construction of highways in Hainan Province, urban areas have become attractive tourist destinations, providing convenient access for tourists both inside and outside the province. Thus, distance from a highway is a crucial factor influencing the distribution of B&Bs in urban areas. However, the relatively limited transportation development in rural areas within the demonstration area restricted the growth of B&Bs [44]. This disparity reflects the prevailing urban–rural development differences in China [45]. Consequently, road network density, which represents transportation accessibility, emerged as a significant factor affecting the distribution of rural B&Bs. While the influence of transportation location on the distribution of rural B&Bs is comparatively weak from an urban–rural perspective, it is foreseeable that enhancing the level of transportation infrastructure in rural areas can foster the development of rural B&Bs.

The development of the urban tourism accommodation industry in the demonstration area preceded that in rural areas, resulting in hotels becoming prime businesses. This spatial spillover effect attracts the clustering of B&Bs in urban areas [46]. Housing density also influences the distribution of B&Bs, possibly because areas with higher neighborhood densities are more likely to have landlords who open B&Bs. The higher neighborhood residential density in rural areas suggests a more concentrated population and potentially

more customer groups and room availability, which positively impacts the development of rural B&Bs.

Moreover, urban areas exhibit a higher supply of leisure and recreational services [47], consequently, urban B&Bs tend to cluster in these areas. In contrast, tourists in rural areas can enjoy a tranquil country experience in the B&Bs and enrich their travel through various leisure and recreational activities. This perspective on urban–rural differences is critical for understanding the influence of amenities on the distribution of B&Bs. Both urban and rural B&Bs are associated with population agglomerations and tend to have high concentrations of amenities. Specifically, tourism-related B&Bs are influenced by leisure and entertainment amenities, providing further evidence of their impact on B&B distribution. A higher density of recreational and entertainment facilities enhances the attractiveness of an area for tourists, thus promoting the development of B&Bs.

Finally, the variance in attraction dependence between urban and rural areas shapes the spatial distribution of B&Bs. In urban areas, significant tourist attractions attract a large number of tourists, leading to higher demand for supporting tourism services, such as transportation, public facilities, and recreational amenities. Urban areas typically have better supporting conditions, resulting in high B&B concentrations in Haikou City, which houses several prominent tourist attractions. However, urban B&Bs are usually located far from national A-class scenic spots. This observation may be attributed to the relatively higher cost of hotels near these scenic spots, making urban B&Bs a more affordable and sought-after accommodation option [48]. Conversely, tourism resources in rural areas require further development to offer experiences distinct from those of urban tourism [49]. Consequently, rural B&Bs tend to cluster in areas with high tourist attractions because a high density of such attractions attracts more tourists, increases area visibility and reputation, and fosters rural B&B development.

## 6. Conclusions and Policy Implications

### 6.1. Conclusions

The tourism B&B industry, being a service-oriented sector for the urban population, exhibits noticeable urban–rural disparities, yet only limited research has explored this feature. Thus, this study investigates the urban–rural spatial distribution characteristics and influential factors of B&Bs within one of China's All-for-one Tourism Demonstration Areas. The findings revealed a significant clustering distribution of B&Bs within regional tourism demonstration areas, with notable disparities in clustering levels between urban and rural areas. Urban areas witness B&B concentration primarily in downtown Haikou, forming an urban B&B agglomeration centered around the city. In contrast, rural B&Bs appear more scattered, predominantly in nodal regions within the demonstration area, characterized by concentrated tourist attractions, high transportation accessibility to cities, a favorable B&B development foundation, and robust living services.

This study further analyzes the primary factors influencing the spatial distribution of urban and rural B&Bs by utilizing exploratory spatial data analysis and ridge regression models. These results indicate that tourism attraction dependence is the most crucial driving force influencing B&B clustering. In urban areas, B&Bs tend to cluster in regions with easy transportation access, favorable tourism locations, and a high density of hotels, owing to the developed economy, robust tourism development, and ample supply of new housing options. Conversely, rural areas display a lower level of tourism resource development and varying internal transportation infrastructure, with neighborhood housing serving as a pivotal source of new accommodations. Consequently, rural B&Bs tend to cluster in nodal areas with high accessibility to urban transportation and dense neighborhood housing. However, the impact of rural tourism destination construction, such as beautiful countryside initiatives, on the spatial aggregation of rural B&Bs remains limited, potentially due to current initiatives that lack long-term consumer retention appeal.

Finally, this study has certain limitations that require further improvements in future research. Specifically, comprehensive B&B data from additional platforms are necessary

to analyze the spatial clustering characteristics of B&Bs in the demonstration area more thoroughly. Moreover, a deeper exploration of other significant factors influencing the clustering of urban and rural B&Bs in the demonstration areas is warranted. These research efforts facilitate a deeper understanding of the development and spatial distribution of urban and rural tourism in China.

*6.2. Policy Implications*

Based on the findings of this study, we propose development strategies aimed at optimizing the spatial structure of B&Bs in All-for-one Tourism Demonstration Areas, focusing on three perspectives: creating tourism destinations, meeting tourists' needs, and fostering the development of the B&B industry. These strategies provide valuable insights for the development of All-for-one Tourism Demonstration Areas and the B&B sector in regions with advantageous tourism resources.

First, the development of All-for-one Tourism Demonstration Areas and B&B service circles should be strategically aligned with regional tourism resources and characteristics to establish a comprehensive regional tourism destination system. Urban areas, leveraging their economic strength and establishing tourism service infrastructure, can focus on developing large-scale scenic spots and urban-oriented B&Bs. By enhancing B&B services, they can offer tourists high-quality accommodation experiences. Meanwhile, rural areas, despite having scattered and underdeveloped tourism resources, have the potential to capitalize on the burgeoning trends of leisure agriculture and rural tourism, which are gaining popularity under the integrated urban–rural development framework. Unique local cultures and vernacular settlements have begun to attract urban middle-class tourists, leading to the incipient formation of tourist B&Bs in rural areas with quality tourism resources. Therefore, rural areas can capitalize on their distinctive local culture and natural assets to promote rural tourism and leisure agriculture. Developing high-quality rural B&Bs can further enhance rural tourism.

Second, tourists' accommodation choices in B&Bs are influenced not only by their travel needs, but also by the B&B's transportation accessibility and the level of surrounding amenities, all aimed at enhancing the overall touring experience. In urban B&Bs, transportation convenience, availability of amenities, and access to recreational opportunities are crucial factors attracting tourists. Conversely, rural B&Bs, constrained by local economic development, appeal to tourists with the convenience of transportation to and from the city as well as the cleanliness and comfort of their accommodations. Therefore, it is vital to consider the diverse needs of tourists when developing and constructing B&Bs in All-for-one Tourism Demonstration Areas. In urban areas, efforts can be made to strengthen public transportation and urban planning to improve transportation convenience and the availability of facilities near urban B&Bs. In rural areas, emphasis should be placed on enhancing the tourism transportation infrastructure to establish seamless connections between urban and rural networks. Additionally, the development of leisure and entertainment facilities near tourist attractions should be prioritized along with the establishment of a well-planned tourism rest system to provide tourists with an enhanced tour experience.

Finally, recognizing the pivotal role of the B&B industry's developmental foundation in driving B&B clustering is essential. In urban areas, where the hospitality industry is well-established and closely linked to tourism, B&Bs are often distributed around prime locations near tourist attractions. In such cases, it is crucial to consider the competitive relationship between the B&Bs and the hotel industry to avoid excessive competition. Conversely, in rural areas, owing to the limited supply of accommodation and the relatively underdeveloped tourist accommodation sector, ensuring the provision of high-quality B&Bs and creating clusters of premium tourist B&Bs are imperative. Moreover, attention should be paid to elevating the development level of the rural tourism accommodation industry to enhance its appeal and cater to the diverse needs of tourists seeking accommodation. The construction of B&Bs in All-for-one Tourism Demonstration Areas necessitates

differentiated policy measures based on the unique conditions of each region, ensuring the healthy and orderly development of the B&B industry.

**Author Contributions:** Conceptualization, L.C.; methodology, A.S. and L.C.; software, A.S.; validation, A.S. and L.C.; formal analysis, A.S.; data curation, A.S.; writing—original draft preparation, A.S.; writing—review and editing, L.C., K.Y. and M.Q.; visualization, A.S.; supervision, L.C.; project administration, L.C.; funding acquisition, L.C. All authors have read and agreed to the published version of the manuscript.

**Funding:** This research was funded by the National Natural Science Foundation of China, grant no. 42271207, and the National Natural Science Foundation of China (Key Program), grant no. 42230705.

**Data Availability Statement:** Not applicable.

**Conflicts of Interest:** The authors declare no conflict of interest.

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
