# Peer review of "Spatial Patterns and Determinants of Bed and Breakfasts in the All-for-One Tourism Demonstration Area of China: A Perspective on Urban–Rural Differences"

_land, doi:10.3390/land12091720_

Round 1

Reviewer 1 Report

I ask that the authors specifically address each of my comments in their responses. The authors did a great job, but the article needs some improvements to be ready for publishment in high-quality journals. I hope my comments, observations, and suggestions will allow the authors to improve the manuscript and work towards publication. Below, I include comments pointing toward some of the issues.

- Introduction and Theoretical Background:

The authors are encouraged to expound upon the concept of B&B as an accommodation style, elucidating its historical development both in a general context and specifically within China.

- Figure (1):

Despite the authors' efforts to clarify the essence of this figure, its complexity might prove perplexing to a general reader. I recommend that the authors restructure the explanations within the figure, employing a format aligned with the three central pillars: "urban-rural perspective," "global perspective," and "spatial distribution." In essence, this would entail reorganizing the existing explanations with added granularity and simplified concepts.

- Study Area:

The authors underscore the uniqueness of their research in investigating B&B in urban-rural areas, as opposed to the conventional focus on larger cities prevalent in the literature. However, this uniqueness is not entirely evident within the "Study Area" section. While the authors do mention their classification strategy—identifying "streets" as urban areas and "townships" and "towns" as rural areas, following reference [31]—greater clarity is warranted. Notably, the referenced work discusses three administrative strategies in China: "city administering county," "converting counties to cities," and "annexation of suburban counties," each yielding distinct administrative boundaries. Consequently, the authors should emphasize this issue more explicitly in their research, ensuring precision and coherence, given its foundational significance to the article's novelty and originality.

- Table (1):

A minor correction required is the repetition of the table header.

- Variable Selection:

The authors' elucidation of the chosen variables is well-structured and comprehensive. However, a more in-depth justification regarding the specific number of selected variables (10 variables) is needed. This reasoning is crucial, particularly considering the intricate nature of the investigated subject matter (B&B) within complex spatial contexts (urban-rural areas), which may need more variables than these selected 10 ones. It is essential that this rationale is substantiated to ensure clarity in the selection process.

In summary, while the authors have displayed significant dedication in their research, addressing the aforementioned points will greatly contribute to the manuscript's overall refinement. Your attention to these aspects will undoubtedly elevate the manuscript's scholarly quality, enhancing its potential for publication.

Author Response

Dear Reviewer 1,

Thank you for your valuable comments.

Reviewer 2 Report

The paper titled Spatial Patterns and Determinants of Bed and Breakfasts in the All-for-one Tourism Demonstration Area of China: A Perspective on Urban-rural Differences represents a major contribution to the field of spatial patterns of B&B development and their influencing factors from the perspective of urban-rural differences. The authors conducted a comprehensive case study in an all-for-one tourism demonstration area in Hainan Province in China, using geospatial analysis methods and ridge regression models. They investigated the characteristics of urban-rural disparities in B&B distribution and defined the primary factors influencing their spatial arrangement. Based on the results, they proposed development strategies for optimizing B&Bs’ spatial structure in the tourism demonstration area and outlined a blueprint for fostering integrated urban-rural development.

This topic is important, original, and relevant in this field of research.

The methodology used is adequate. Also, the quotations are relevant, and the references are appropriate. The research design is appropriate, and the methods are adequately described. The results are presented adequately and comprehensibly. All figures and tables are well-presented and clear.

The conclusion is very detailed and supported by the results, and they are consistent with the evidence and arguments presented in the manuscript.

Some technical issues:

In some places where a literary source is cited in parentheses, it is necessary to make a space between the word and the open parenthesis (for example technology [1, 2] instead of technology[1, 2].

The article is acceptable for publication in Land after minor revision, in the special issue "Mega-City Regions in the Global South”, section: Urban Contexts and Urban-Rural Interactions.

Author Response

Dear Reviewer,

Thank you for your valuable comments.
